# MCD Diet Modulates HuR and Oxidative Stress-Related HuR Targets in Rats

**DOI:** 10.3390/ijms24129808

**Published:** 2023-06-06

**Authors:** Andrea Ferrigno, Lucrezia Irene Maria Campagnoli, Annalisa Barbieri, Nicoletta Marchesi, Alessia Pascale, Anna Cleta Croce, Mariapia Vairetti, Laura Giuseppina Di Pasqua

**Affiliations:** 1Unit of Cellular and Molecular Pharmacology and Toxicology, Department of Internal Medicine and Therapeutics, University of Pavia, 27100 Pavia, Italy; andrea.ferrigno@unipv.it (A.F.); lauragiuseppin.dipasqua01@universitadipavia.it (L.G.D.P.); 2Interuniversity Center for the Promotion of the 3Rs Principles in Teaching and Research (Centro 3R), 56122 Pisa, Italy; 3Unit of Pharmacology, Department of Drug Sciences, University of Pavia, 27100 Pavia, Italy; lucreziairenem.campagnoli01@universitadipavia.it (L.I.M.C.); nicoletta.marchesi@unipv.it (N.M.); alessia.pascale@unipv.it (A.P.); 4IGM-CNR, Unit of Histochemistry and Cytometry, University of Pavia, 27100 Pavia, Italy; annacleta.croce@igm.cnr.it

**Keywords:** HuR, MnSOD, HO-1, NAFLD, NASH, MCD, liver

## Abstract

The endogenous antioxidant defense plays a big part in the pathogenesis of non-alcoholic fatty liver disease (NAFLD), a common metabolic disorder that can lead to serious complications such as cirrhosis and cancer. HuR, an RNA-binding protein of the ELAV family, controls, among others, the stability of MnSOD and HO-1 mRNA. These two enzymes protect the liver cells from oxidative damage caused by excessive fat accumulation. Our aim was to investigate the expression of HuR and its targets in a methionine-choline deficient (MCD) model of NAFLD. To this aim, we fed male Wistar rats with an MCD diet for 3 and 6 weeks to induce NAFLD; then, we evaluated the expression of HuR, MnSOD, and HO-1. The MCD diet induced fat accumulation, hepatic injury, oxidative stress, and mitochondrial dysfunction. A HuR downregulation was also observed in association with a reduced expression of MnSOD and HO-1. Moreover, the changes in the expression of HuR and its targets were significantly correlated with oxidative stress and mitochondrial injury. Since HuR plays a protective role against oxidative stress, targeting this protein could be a therapeutic strategy to both prevent and counteract NAFLD.

## 1. Introduction

Non-alcoholic fatty liver disease (NAFLD), the hepatic manifestation of metabolic syndrome, characterized by disrupted lipid metabolism, fat accumulation, and lipid droplet formation, affects 20–30% of the human general population, rising up to 75% in obese or diabetic people [1]. NAFLD ranges from simple steatosis to non-alcoholic steatohepatitis (NASH), a fibrotic and chronic inflammatory state occurring in about 10% of individuals with NAFLD, and is potentially able to progress toward cirrhosis and hepatocellular carcinoma (HCC) [2].

Hepatic fibrosis is the main factor behind liver-related mortality in patients with NASH [3]. According to the multiple-hit hypothesis, the pathogenesis of NAFLD is a multifactorial process [4] and different factors concur with its development and progression, including oxidative stress, activation of inflammatory pathways [5], mitochondrial dysfunction [6], endoplasmic reticulum stress [7], nutritional habits [8], insulin resistance [9], altered gut microbiota [10], and genetic/epigenetic factors [11]. Around 2% of cases are drug-related and classified as drug-induced steatohepatitis (DISH) [12]. 

Oxidative stress is considered one of the pivotal processes behind the transition from simple steatosis to NASH; in fact, reactive oxygen species (ROS) directly and indirectly contribute to the up-regulation of pro-inflammatory cytokines and chronic inflammatory response, together with the activation of hepatic stellate cells (HSC) and the development of hepatic fibrosis [13,14]. Multiple factors contribute to oxidative stress; some of them promote the increase in pro-oxidant activities, while others contribute to the decrease of the antioxidant function [15]. Many pieces of evidence suggest that mitochondrial DNA abnormalities are responsible for the increase of oxidative stress in NASH [14]. In mice fed with a high-fat diet, reduced stability of oxidative phosphorylation subunits was observed in association with an impairment in complex III and F0F1-ATP synthase activities, resulting in ATP deficiency and increased ROS production [16]. Similar alterations were also found in patients with alcoholic fatty liver disease (AFLD), leading to electron leakage and increased production of superoxide and hydrogen peroxide [17]. Low levels of glutathione (GSH), GSH peroxidase, manganese-dependent superoxide dismutase (MnSOD), and catalase were also found in NAFLD animal models and NASH patients, as well as increased activity of CYP2E1, which is an important microsomal source of oxidative stress [18]. Inhibition of heme oxygenase-1 (HO-1) and nuclear factor erythroid 2-related factor 2 (NRF2) also contributes to the general reduction in the antioxidant defense [15]. In the development of NASH, oxidative stress also occurs because of lipid peroxidation by specific polyunsaturated fatty acids (PUFAs), along with the formation of highly reactive aldehyde products, such as malondialdehyde (MDA) and 4-hydroxy-2-non-enal (4-HNE) [19], which are the most studied and are also known for their ability to directly or indirectly induce hepatic inflammation [20]. In fact, oxidized phospholipids have been found to bind to certain toll-like receptors (TLRs), thereby mediating pro-inflammatory signals. For example, the administration of malondialdehyde/albumin adducts in sinusoidal endothelial cells from rat livers induced the secretion of pro-inflammatory mediators such as tumor necrosis factor (TNF)-α and triggered pro-fibrotic responses [21].

Key players in tuning inflammatory and oxidative stress responses are the RNA binding proteins (RBPs), which are able to modulate the expression of the multiple factors involved within these processes at a post-transcriptional level. Specifically, the RBPs embryonic lethal abnormal vision (ELAV) proteins contribute to the regulation of gene expression by means of their ability to bind to ARE (adenine-uracil-rich elements) sequences present in target mRNAs, mainly increasing their cytoplasmic stability and/or rate of translation [22]. The human antigen R (HuR; a.k.a. ELAVL-1/ELAV), is an ubiquitously expressed RBP that has been implicated in a plethora of conditions, including cancer [23], atherosclerosis [24], pulmonary fibrosis [25], and neurological disorders [26,27]. Recently, several studies have also considered its role in NAFLD, discovering that HuR exerts a protective role by targeting glucose and lipid metabolism [28,29]. Of note, HuR liver-specific deletion (HuR^hKO^) was also found to promote the transition from simple steatosis to NASH in mice fed with a high-fat diet through the activation of the fibrosis/HSC pathway and the inhibition of the farnesoid X receptor (FXR) pathway [28]. In addition, expression levels of key genes involved in inflammatory and stress responses were significantly increased in HuR^hKO^ mice [30].

HuR was found to also be involved in the modulation of oxidative stress by targeting antioxidant enzymes, such as MnSOD and HO-1 [31,32,33,34,35]. However, no information is currently available about the expression of HuR and HuR-targeted antioxidant enzymes in an animal model characterized by elevated hepatic oxidative stress. This is the case of rats administered with the MCD diet, which causes hepatic steatosis, liver injury, and inflammation and triggers the fibrotic process, producing a spectrum of changes that mimics NASH [36]. 

In this work, we used the MCD diet rat model to investigate the regulation of HuR and its targets MnSOD and HO-1 in response to oxidative stress injury as possible factors taking part in the development of NASH. 

## 2. Results

In this study, five-week-old male Wistar rats fed by the MCD or an isocaloric control diet (CTRL) for 3 weeks or 6 weeks were used.

### 2.1. Model Characterization

#### 2.1.1. Enzyme Release after MCD Diet Administration

The liver injury was determined in terms of alanine aminotransferase (ALT) and aspartate aminotransferase (AST), the enzyme transaminases released by the liver in the blood. A significant increase in ALT and AST levels was observed in MCD as compared with the CTRL group at both times of diet treatment (Table 1).

#### 2.1.2. MCD Diet Administration Increases Oxidative Stress and Mitochondrial Dysfunction

The rate of lipid peroxidation was quantified using malondialdehyde (MDA), the prototype of thiobarbituric acid reactive substances (TBARS) and one of the most frequently measured biomarkers of oxidative stress [37]. MDA formation was evaluated in hepatic tissue from rats fed with a control (CTRL) diet and MCD diet for 3 and 6 weeks. The graph shows a significant increase in MDA formation in MCD groups, after both 3 and 6 weeks, when compared with their respective CTRL groups. In addition, the content of MDA is significantly augmented in the hepatic tissue of rats fed with the MCD diet for 6 weeks compared with the animals receiving only 3 weeks of the MCD diet (Figure 1a). 

A significant increase in ROS was detected in MCD rats after 6 weeks of diet when compared with the CTRL group; the same trend was observed also after 3 weeks of diet, although the difference was not significant (Figure 1b). Comparable results were found when mitochondrial ROS were taken into account: a significant increase in ROS production was observed both after 3 and 6 weeks of MCD diet administration in hepatic fresh isolated mitochondria with respect to their relative controls (Figure 1c).

A mitochondrial dysfunction in MCD-treated rats was also confirmed by the reduced ratio between the State3/State4 of mitochondrial respiration. As expected, a reduction in mitochondrial respiration rate was already noticeable after 3 weeks of the MCD diet and it decreased significantly after 6 weeks of treatment (Figure 1d). 

#### 2.1.3. MCD Diet Administration Promotes Hepatic Lipid Accumulation

Liver sections from CTRL rats and MCD rats after 6 weeks of diet administration were stained with hematoxylin and eosin (H&E), allowing to appreciate an abundant deposition of lipids in the hepatic tissue of MCD rats when compared with the CTRL group (Figure 2a,b). The evaluation of the total lipid content by using the fluorescent dye Nile Red confirmed a significant increase in the MCD group with respect to the CTRL group (Figure 2c). The same significant trend was found in the number and area of lipid droplets obtained after ImageJ analysis on H&E liver sections (Figure 2d,e).

#### 2.1.4. MCD Diet Administration Reduces Hepatic ATP Content and NADPH Bound/Free Ratio

Given that the respiratory ratio displayed a significant decrease in mitochondrial respiration after 6 weeks of MCD diet administration, the ATP content was evaluated in the hepatic tissue from rats fed with the CTRL and MCD diet for 6 weeks. As expected, a significant reduction in ATP content was found in MCD-treated rats when compared with CTRL rats, confirming again that the MCD diet administration promotes oxidative stress and mitochondrial dysfunction (Figure 3a).

A significant decrease in NADPH bound/free ratio was also found in 6-week MCD-treated rats compared with the CTRL group (Figure 3b). The ratio between bound and free NAD(P)H, here used as a parameter related to mitochondrial functionality, was calculated from the liver autofluorescence spectra following a curve fitting analysis. This procedure allows us to estimate the relative contribution of NAD(P)H in the bound and free forms to the overall emission area taking advantage of their different spectral position. The relative area values of the two emission bands typical of bound and free NAD(P)H can then be used to calculate the NAD(P)H bound/free ratio. 

### 2.2. HuR and Its Targets MnSOD and HO-1

#### 2.2.1. Effect of the MCD Diet on HuR Protein Expression

The expression of HuR was quantified in the hepatic tissue of rats fed with CTRL and MCD diet for 3 and 6 weeks. Western blotting analysis showed a significant decrease in HuR protein level in the liver of rats after 6 weeks of the MCD diet compared to the relative CTRL group; after 3 weeks of diet, a slight although not significant protein reduction was detected (Figure 4). 

#### 2.2.2. Effect of the MCD Diet on MnSOD and HO-1 Protein Expression

The expression of the HuR targets MnSOD and HO-1 was quantified in hepatic tissue from rats fed with the CTRL and MCD diet for 3 and 6 weeks. Western blotting analysis revealed that both MnSOD (Figure 5a) and HO-1 (Figure 5b) protein expression was significantly different between CTRL and MCD rats, showing a similar trend. In more detail, a lower protein content was found in the liver of both MCD groups (after 3 and 6 weeks) compared to the relative CTRL groups, with an even more evident reduction following 6 weeks. 

#### 2.2.3. Correlation between HuR, Oxidative Stress, and HuR Targets MnSOD and HO-1

Analysis of correlations between HuR and oxidative stress indexes was performed. A significant inverse correlation was found between HuR and TBARS and HuR and ROS production in the whole tissue, while a significant positive correlation was observed between HuR and the NADPH bound/free ratio, thus confirming the protective role of HuR in this NAFLD model. Similar correlations were also found when considering the HuR targets MnSOD and HO-1 versus TBARS, ROS, and NADPH (Figure 6).

## 3. Discussion

Some of the most common chronic liver diseases are caused by NAFLD, a condition affecting one-fourth of the current adult population, according to a recent epidemiologic meta-analysis [38]. Effective interventions to prevent the progression from simple steatosis to NASH are urgently needed.

The choice of an appropriate animal model is essential for developing a treatment for this rapidly growing disease. There are different diet-based models of NAFLD, including, among others, the MCD diet, the high-fat (HF) diet, the high-fat, high-fructose (HFF) diet, and the cholesterol and cholate (CC) diet. In MCD animals, choline deficiency causes an abrupt cessation of phosphatidylcholine de novo synthesis and triglyceride liver export of very low-density lipoproteins (VLDL), resulting in intrahepatic lipid accumulation [39]. In addition, the lack of methionine, an essential intermediate in GSH synthesis, causes the overload of the antioxidant system resulting in a higher susceptibility to oxidative stress injury [40]. In animals fed with an MCD diet, the typical NASH manifestations, such as macrosteatosis, oxidative stress, chronic inflammation, and fibrosis, occur in the absence of systemic insulin resistance. In fact, the MCD diet is more suitable for studying the chronic inflammatory state and collagen deposition [41]. HF diet induces metabolic syndrome and simple steatosis, usually not spontaneously progressing toward NASH unless an additional stimulus is provided [42]. The HFF diet induces liver pathology prior to, or even in the absence of, obesity or insulin resistance in association with skeletal muscle extracellular lipid accumulation, inflammation, and autophagy [43] and mild oxidative stress [44]. The CC diet was developed to mimic the cholesterol-mediated insulin resistance, cardiovascular risk increase, and NASH; the CC diet produces steatosis, inflammation, HSC activation, and fibrosis; however, animals fed with this diet are not insulin-resistant; and tend to lose weight and have lower triglyceride levels, with respect to standard chow-fed mice [2]. 

As a multifactorial process, NAFLD develops in response to many concurring factors, including genetic/epigenetic factors [11], hypercaloric diet [8], insulin resistance [9], oxidative stress and inflammation [5], mitochondrial dysfunction [6], endoplasmic reticulum stress [7], and altered gut microbiota [10]. HuR has already been found to have a role in modulating various of the above-mentioned processes. In mice, HuR hepatocyte-specific knock-out significantly affected the expression of genes involved in the hepatic fibrosis/HSC activation pathway, including the platelet-derived growth factor (PDGF) signaling pathway, the epithelial–mesenchymal transition pathway, and the hepatic fibrosis signaling pathway. A reduced Farnesoid X receptor (FXR)/Retinoid X Receptor (RXR) activation in HuR-deficient hepatocytes was also observed [28]. The FXR pathway regulates lipid homeostasis; in fact, FXR activation increases TAG clearance and blocks sterol regulatory element binding protein 1–mediated lipogenesis; additionally, FXR signaling inhibits HSC activation, resulting also in the protection against fibrogenesis [45].

Liver-specific HuR knockout mice showed exacerbated HFD-induced steatosis, mediated by decreased stability of the HuR target phosphatase and tensin homolog deleted on the chromosome 10 (PTEN) mRNA [46]. PTEN acts downstream of the insulin receptor5; liver-specific PTEN deficiency promotes NAFLD and tumorigenesis while improving glucose tolerance [47]. Finally, adipose-specific HuR knockout mice showed obesity, insulin resistance, glucose intolerance, and hypercholesterolemia attributed to decreased expression of the HuR target adipose triglyceride lipase (ATGL), involved in the control of obesity and metabolic syndrome [29].

The main aim of the present research was to investigate whether MCD diet-induced oxidative stress was associated with changes in the expression of HuR and of two of its targets, namely, MnSOD and HO-1, involved in antioxidant defense mechanisms; consequently, the MCD diet model was used since methionine-choline deficiency is known to induce a dramatic increase in oxidative stress [41]. Wistar rats were fed with MCD and isocaloric control (CTRL) diets and sacrificed after 3 and 6 weeks. As expected, a significant rise in AST and ALT release was detected in MCD animals; in particular, after 6 weeks, a 4x increase in ALT and a 2x increase in AST were detected in comparison to CTRL rats, which is similar to what other researchers found in an analogous model [48]. After 6 weeks, the hepatic lipid content increased dramatically, as confirmed by the Nile Red fluorescence assay and by the lipid droplet count and droplet area on H&E liver sections. Macrovesicular steatosis was also observed, as previously described by others [49,50]. Saturated free fatty acids accumulating in MCD-fed rodents can cause lipotoxicity through c-Jun N-terminal kinase-1 (JNK-1) activation, leading to mitochondrial damage and ROS production [51]. Indeed, we found an increase in tissue ROS production and lipid peroxidation, the latter increasing in a time-dependent manner. In parallel, ROS production in isolated mitochondria also increased significantly after 3 and 6 weeks of MCD diet administration, resulting in a dramatic reduction in mitochondrial efficiency, as suggested by the 4x drop in the respiratory control ratio in the mitochondria isolated from 6-week MCD rats. The curve-fitting analysis of the liver autofluorescence spectra also revealed a decrease in the NADPH bound/free ratio in 6-week MCD livers. Since the NADPH bound/free ratio reflects the real coenzyme involvement in the redox metabolism [52], the finding is consistent with an impairment of the engagement of MCD livers in aerobic respiration, as supported by biochemical data on the decrease in liver ATP levels. 

In the context of a well-characterized NASH model in rats, we evaluated the changes in the expression of HuR and its targets. After 6 weeks of MCD diet administration, HuR levels were significantly lower in comparison with CTRL diet-fed rats, as well as MnSOD and HO-1. In addition, HuR, MnSOD, and HO-1 levels were significantly correlated with ROS, TBARS (negative correlation), and NADPH free/bound ratio (positive correlation), thus indicating a protective role played by HuR and its targets in this NAFLD model. 

HuR’s role in NAFLD development is currently being investigated only in HF-induced murine models of NAFLD. In a recent study, HuR-mediated modulation of ubiquinol-cytochrome c reductase binding protein (UQCRB), apolipoprotein B-100 (APOB), and NADH dehydrogenase (ubiquinone) 1 beta subcomplex subunit 6 (NDUVB6) was evaluated. HuR was found to stabilize Uqcrb, Ndufb6, and Apob mRNAs; in fact, hepatocyte-specific HuR knockout reduced the expression of the respective protein factors, resulting in a reduction of liver lipid export and ATP production and in the aggravation of HF diet-induced NAFLD [53]. In addition, HuR was found to be a repressor of H19 [30], a long non-coding RNA reported to modulate hepatic metabolic homeostasis in NAFLD, whose overexpression is associated with steatosis and the development of obstructive cholestatic liver fibrosis [54]. Currently, no study has been conducted to evaluate the changes in HuR and in HuR-mediated modulation of oxidative stress in an MCD model of NAFLD, although the capability of HuR to modulate the antioxidant system in other organs is already known. In a transcriptome-wide RNA-binding analysis on HeLa cells, ten HuR binding sites were found in the 3′ UTR region of MnSOD mRNA; in fact, HuR knockdown negatively modulated MnSOD mRNA and protein expression [31]. Many studies have recently evaluated HuR-mediated posttranscriptional regulation of MnSOD in non-NASH-related models. For instance, in ovarian cancer cells, HuR-mediated MnSOD mRNA stabilization led to a rapid increase in mitochondrial antioxidant capacity [32]. In human brain microvascular endothelial cells, oxidative stress induced MnSOD downregulation, further accentuated by the administration of HuR inhibitors [33]. Notably, antioxidant enzymes such as MnSOD, catalase, and GSH peroxidase, have a pivotal role in reducing anion radicals generated within mitochondria, contrasting the progression from simple steatosis to NASH [55]. In fact, sirtuin 3 (SIRT3) KO mice, which exhibit decreased MnSOD activity and increased oxidative stress when subjected to an MCD diet, present worsened NASH symptoms in comparison with wild-type mice [56]. Moreover, MnSOD levels seem to be lower in NASH patients [14], and MnSOD functional genetic variations were found to be associated with a higher risk of NASH in a human population [55]. HO-1 also has a role in NASH pathogenesis. Recently, it was found that the administration of the HO-1 inducer hemin significantly ameliorates the severity of steatosis, inflammation, and fibrosis and also decreases the serum ALT and AST levels by inhibiting the activation of canonical and noncanonical Wnt signaling pathways in MCD mice [57]. Furthermore, a dramatic decrease in hepatic ROS, in association with a reduction in endoplasmic reticulum stress, was observed in mice fed with an MCD diet when treated with hemin [58]. Similarly to MnSOD, HO-1 expression also is HuR-dependent [35]. In renal tubular cells, HuR-binding regions were found in the 3′-UTR region of HO-1 mRNA [59]; moreover, HuR silencing downregulates HO-1, leading to increased hepatotoxicity in an in vitro model of ischemia-reperfusion [60]. 

Our results suggest for the first time that, in an MCD model of NASH, HuR levels are downregulated, and that the HuR targets MnSOD and HO-1 are downregulated as well, thus suggesting that a defective HuR-mediated modulation of antioxidant enzymes may contribute to the progression from simple steatosis to NASH. The present work has limitations; here we described how NASH development affects the expression of HuR and its targets, but the effects of HuR positive or negative modulation on NASH development have not been evaluated here. Consequently, in the future development of this work, we will evaluate the effect of HuR knock-out on an MCD model of NASH. However, even though further studies are needed to confirm our findings, the present results underscore HuR as a promising pharmacological target in the treatment of NASH. 

## 4. Materials and Methods

### 4.1. Animal Model and Experimental Procedures

The animal model employed in this study was approved by the Italian Ministry of Health and the Pavia University Animal Care Commission (Authorization number 163/2020). Five-week-old male Wistar rats (Charles River Laboratories, Calco, Italy) were used in this study. After a 1-week quarantine period, animals were fed with the MCD diet (Laboratorio Dottori Piccioni, Gessate, Italy) or isocaloric control diet (CTRL) for 3 weeks (3w; n = 7 for each group) or 6 weeks (6w; n = 15 MCD group; n = 6 CTRL group), as previously described [61]. At the end of diet treatment, rats were anesthetized with sodium pentobarbital injected intraperitoneally (40 mg/kg), the abdomen was opened, and blood was collected from the inferior caval vein. The liver biopsies from the median and right lobes were collected and snap-frozen in liquid nitrogen for further analyses, or fixed in formalin for paraplast inclusion, while the left lobe was used for fresh mitochondria isolation.

### 4.2. Materials

The MCD diet was purchased by Laboratorio Dottori Piccioni (Gessate, Italy). All chemicals, when not specified, were purchased from Sigma-Aldrich (Milano, Italy).

### 4.3. Blood Sample Preparation and Enzyme Evaluation

Blood samples were obtained from the inferior caval vein, and samples were stored in tubes containing Na_2_EDTA used as an anticoagulant. After centrifuging blood samples at 1520× *g* for 15 min, RT, plasma was obtained, and aliquots were sent to MyLav–Laboratorio La Vallonea (Passirana di Rho, Italy) for ALT and AST evaluation by kinetic UV method.

### 4.4. Mitochondria Isolation

The hepatic left lobe was used for mitochondria isolation according to the Lehninger et al. method using a differential centrifugation [62]. Briefly, the hepatic left lobe was immediately washed in 100 mL of ice-cold homogenization medium (0.25 M sucrose, 1 mM EDTA, 5 mM HEPES; pH 7.2). Afterward, the tissue was weighed and cut into small pieces, and a volume equal to twice the weight of the hepatic left lobe was added to homogenate it using a Teflon/glass Potter homogenizer (Sartorius, Gottingen, Germany). The homogenate was filtered through a double gauze and then centrifuged at 500× *g* for 10 min. The collected supernatant was centrifuged again at 9400× *g* for 10 min and the obtained pellet was resuspended in a free-EDTA buffer (0.25 M sucrose, 5 mM HEPES) and centrifuged again for 10 min at 10,000× *g*. Single mitochondrial preparations were obtained for each individual animal (MCD 3w and CTRL 3w, n = 7; MCD 6w, n = 15 and CTRL 6w, n = 6), and the protein concentration was determined using the Lowry method [63]. Isolated mitochondria were kept on ice and used immediately for mitochondrial respiratory control index (RCI) determination and ROS evaluation or frozen in liquid nitrogen for further analyses.

### 4.5. Respiratory Control Index Evaluation

RCI was assayed by means of a Clarke electrode in a sealed chamber as previously described [64]. Shortly, mitochondrial O_2_ consumption was measured at 25 °C in a sealed chamber by a Clark-type electrode, adding freshly isolated mitochondria (1 mg/mL) to 2 mL of respiration buffer (5 mM Mg^2+^ and 0.5 mM EGTA). After a stabilization phase, mitochondrial respiration was initiated adding 10 mM succinate in combination with 1 μM rotenone, while oxidative phosphorylation was triggered with 0.5 mM adenosine diphosphate (ADP) administration. The calculation of RCI, expressed as the ratio between State 3 (the highest respiratory rate reached in the presence of respiratory substrates and ADP) and State 4 (the lowest respiratory rate obtained when respiratory substrates are present, but ADP is completely consumed), was obtained recording O_2_ consumption.

### 4.6. Mitochondrial ROS Production

Mitochondrial ROS production was assayed incubating freshly isolated mitochondria (1 mg/mL) in the dark with 5 µM dichlorodihydrofluorescein diacetate (DCFH-DA) for 10 min. At the end of incubation, the mitochondrial suspension was centrifuged at 660× *g* for 10 min at RT, and the obtained pellet was resuspended in the respiration buffer. An amount of 2 mL aliquots of mitochondria (1 mg/mL) was transferred into a UV quartz cuvette, and 10 mM succinate in combination with 1 μM rotenone was added. After the stabilization, 0.5 mM ADP was administered, and the fluorescence intensity (Ex 493/Em 520) was monitored for 10 min using a fluorescence spectrometer LS 50 B (Perkin Elmer Inc., Waltham, MA, USA). Data are reported as arbitrary units (A.U.) (State 3 DCFH fluorescence % increase).

### 4.7. Oxidative Stress and ATP Content Evaluation

The extent of lipid peroxidation was determined as thiobarbituric acid reactive substances (TBARS) formation in liver homogenates, according to the method of Esterbauer and Cheeseman [65]. Malondialdehyde (MDA) was used as the standard to calculate the TBARS concentration. ROS production in hepatic tissue was quantified by the DCFH-DA method based on the ROS-dependent oxidation of DCFH to dichlorofluorescein, as already mentioned [66]. ROS are expressed as arbitrary units (A.U.). ATP levels were measured by means of a luciferin-luciferase kit (Perkin Elmer Inc., Waltham, MA, USA) and expressed as nmol/mg proteins. Luminescence was evaluated on a Perkin Elmer Wallac Victor^2^ multilabel counter, using a white 96-well plate [64].

### 4.8. NADPH Bound/Free Ratio Evaluation

The autofluorescence spectra were recorded from unfixed liver tissue sections cut at the cryostat by means of a microspectrograph (Leitz, Wetzlar, Germany), operating under epi-illumination and using a 100 W/Hg lamp (Osram, Berlin, Germany) as the light excitation source. Excitation light was selected by means of a 366 nm interference filter (FWHM = 10 nm). Fluorescence signals were collected through a 50/50 dichroic mirror and a 390 nm long pass filter, with a 40× objective (n.a. 0.75), and driven to the detection system (multichannel analyzer, Hamamatzu PMA-12, Hamamatsu Photonics Italia Srl, Arese, Italy) via an optically coupled fiber optic probe. The spectra recorded in the 390–700 nm interval were submitted to the curve-fitting procedure (PeakFit: SPSS Science, Chicago, IL, USA) based on the Marquardt algorithm [67] and previously described in detail [52]. Briefly, the contribution of each hepatic endogenous fluorophore to the overall emission was estimated using the half-Gaussian Modified Gaussian (GMG) functions, defined by the spectral parameters describing the profile of each hepatic endogenous fluorophores according to the center peak position (λ) and full width at the half maximum (FWHM) parameters. The GMG functions were combined and submitted to subsequent adjustments to achieve the best fit between the profile of the sum of the contribution bands of the fluorophores and the measured spectrum. The goodness of the fitting depends on the minimization of the minimum squared error results and was assessed by means of the analysis of residuals and the coefficient of determination (r^2^). Before the fitting procedure, the measured spectra were normalized to 100 A.U. at the maximum peak. The contribution of each fluorophore was estimated as the fraction of the overall spectral area, and the area values of the bands ascribed to NAD(PH) bound (λ = 444 nm; FWHM = 105 nm) and NAD(PH) free (λ = 463 nm; FWHM = 115 nm) were used to calculate the NAD(P)H bound/free ratios.

### 4.9. Tissue Histology and Staining

Fresh liver biopsies from rats were fixed in 2% p-formaldehyde in 0.1 M phosphate buffer at pH 7.4 for 24 h and processed routinely until they were embedded in Paraplast wax. Samples were sliced into 8 μm thick sections and stained with hematoxylin–eosin (Sakura Finetek, Mestre, Italy). Slices were then evaluated under a light microscope (Nikon Eclipse E800). Lipid droplet number and area were estimated using ImageJ software version 9.2.0.

### 4.10. Hepatic Lipid Extraction and Quantification

Lipid extraction was performed from hepatic frozen biopsies according to Lyn-Cook et al. [68]. Frozen tissues (50–70 mg each) were homogenized in 200 µL of water. Lipids were extracted by adding 1 mL chloroform-methanol (2:1), and samples were incubated for 1 h at RT with intermittent agitation. After centrifuging at 1520× *g* for 5 min at RT, the separated lipid-containing lower fraction was transferred to a clean tube and N_2_-dried. Pellets were re-suspended in 100 µL of 100% ethanol. Quantification of the total lipid content was performed as previously described [69]. Briefly, 5 µL of lipid extracts were transferred in a black 96-well plate containing 95 µL of phosphate-buffered saline (PBS) and Nile Red (1 mg/mL in DMSO). Fluorescence intensity was measured by a Perkin Elmer Wallac Victor^2^ multilabel counter. Results are expressed as arbitrary units (A.U.).

### 4.11. Western Blot

Frozen hepatic tissue was homogenized in an ice-cold lysis buffer supplemented with protease inhibitors (10 µL/mL) and centrifuged at 15,000× *g* for 10 min. The protein content of each sample was measured via Bradford’s method, using bovine serum albumin (BSA; Sigma Aldrich, Darmstadt, Germany) as the internal standard. Proteins were diluted in sodium dodecyl sulfate (SDS) protein gel loading solution 2×, boiled for 5 min at 95 °C, separated by 12% SDS-polyacrylamide gel electrophoresis (PAGE), and then transferred to nitrocellulose membranes. Unspecific sites were blocked with 6% milk in TBST buffer (10 mM Tris-HCl, 100 mM NaCl, 0.1% (*v*/*v*) Tween 20, pH 7.5) at room temperature for 1 h, and then the membranes were incubated with primary antibodies overnight at 4 °C under gentle agitation. The anti-HuR (Santa Cruz Biotechnology, Dallas, TX, USA), the anti-MnSOD (Santa Cruz Biotechnology), and the anti-HO-1 (Santa Cruz Biotechnology) mouse monoclonal antibodies were diluted 1:1000, 1:500, and 1:500, respectively, in TBST buffer containing 6% (*v*/*v*) milk. The next day, the membranes were washed and then incubated with the secondary antibodies for 1 h, based on the datasheet. The membrane signals were detected by chemiluminescence by means of an Imager Amersham 680 detection system using α-tubulin mouse monoclonal antibody (Sigma Aldrich) at 1:1000 to normalize the data. Densitometric analysis of Western blot data was performed using the ImageJ image-processing program. 

### 4.12. Statistical Analysis

The sample size was calculated using a balanced one-way analysis of variance power calculation tool provided with R statistical software (version 4.1.0). The F size value of 0.8 was calculated on the basis of data previously published in the literature; a significance level of 0.05 and a power of 0.8 were used according to the recommendations of the National Ethical Committee. The sample size obtained was rounded up to account for the possibility of animals not meeting the inclusion criteria.

All data were statistically analyzed using R statistical software (version 4.1.0) and RStudio integrated development environment (version 2022.02.3 Build 492).

Statistical analysis was performed with one-way ANOVA with Tukey’s multiple comparison test, as a post hoc test, or Dunn’s test, as appropriate for multiple comparison. When data distribution was not normal and the analysis of variances was not homogeneous, the Games–Howell test was used. To compare only two groups, statistical analysis was performed with Student’s *t* test. When data distribution was not normal, Wilcoxon’s rank test was used. Shapiro’s normality test and Levene’s test were used to assess the normality and homogeneity of variances, respectively. The correlation analysis was performed using Pearson’s test or Spearman’s rank correlation test for parametric and non-parametric data, respectively. The results are expressed as a mean value ± standard error (S.E.M.). The value of *p* = 0.05 was used as the threshold level of significance.

## Figures and Tables

**Figure 1 ijms-24-09808-f001:**
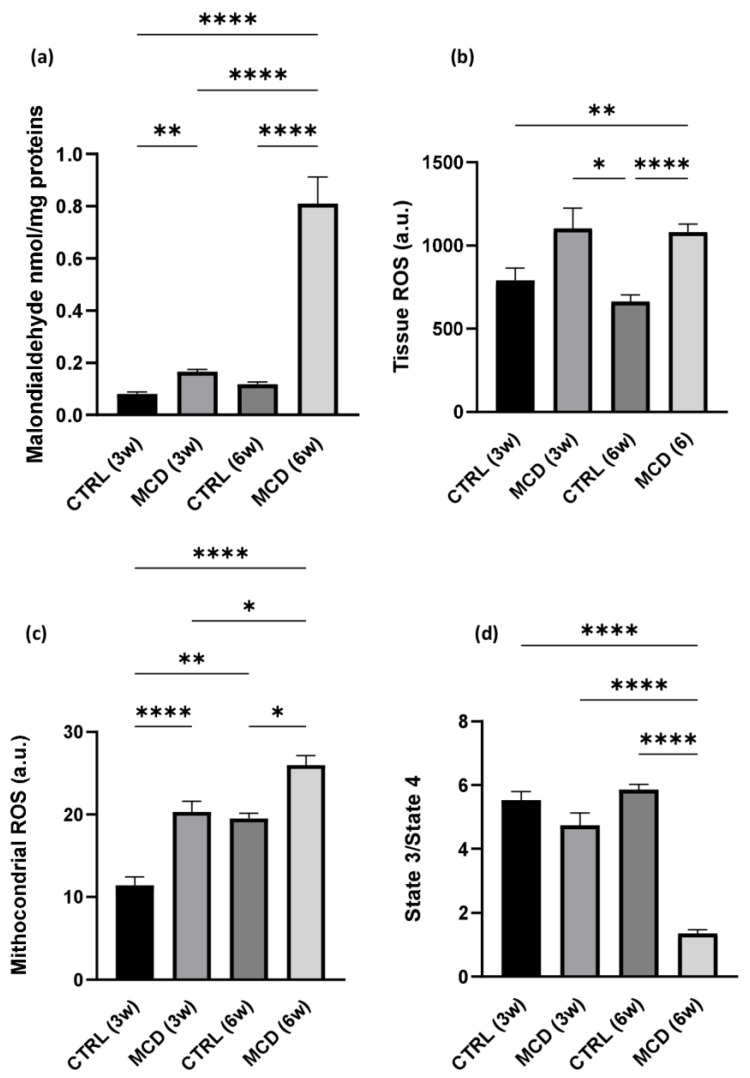
Oxidative stress and mitochondrial dysfunction evaluation after MCD diet administration. Rats were fed with the CTRL diet for 3 weeks (n = 7) and 6 weeks (n = 6) and the MCD diet for 3 weeks (n = 7) and 6 weeks (n = 15). (**a**) TBARS evaluation in hepatic tissue; (**b**) ROS production in hepatic tissue; (**c**) mitochondrial ROS production in fresh isolated hepatic mitochondria; (**d**) State3/State4 ratio as mitochondrial respiration index in fresh isolated hepatic mitochondria. The results are expressed as mean ± S.E.M.; * *p* < 0.05, ** *p* < 0.01, **** *p* < 0.0001; the Games–Howell test was used to evaluate statistical significance. CTRL = isocaloric control diet; MCD = methionine- and choline-deficient diet; 3w = three weeks; 6w = six weeks.

**Figure 2 ijms-24-09808-f002:**
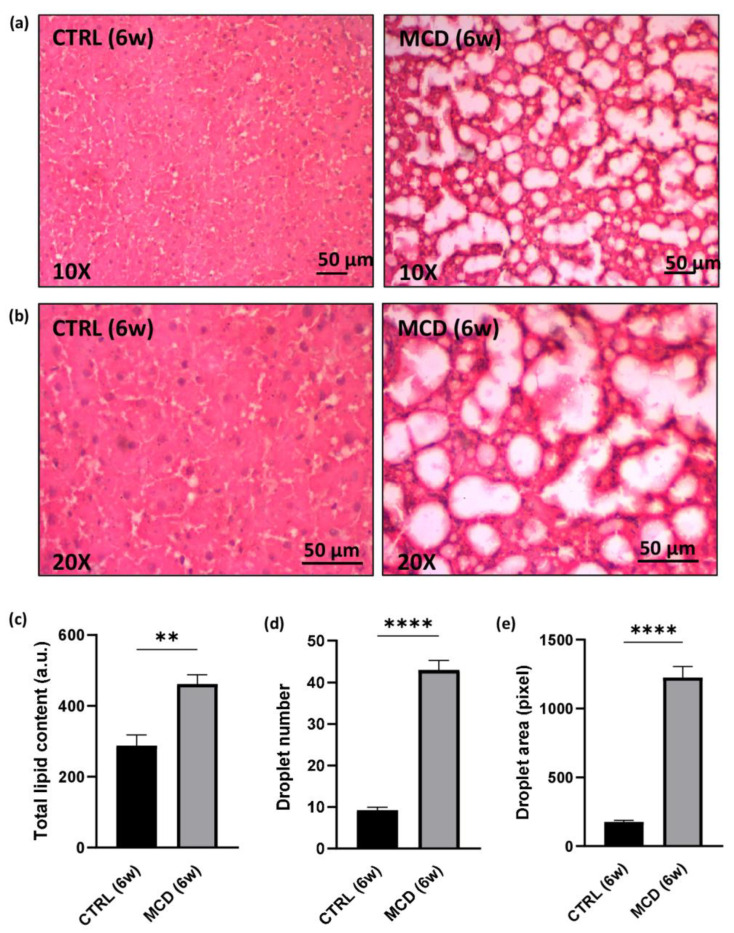
Evaluation of lipid accumulation in 6-week MCD diet-fed and CTRL rats. The animals were fed with the CTRL (n = 6) and MCD (n = 15) diet for 6 weeks. (**a**,**b**) Hematoxylin and Eosin stained liver sections. Panels at 10× (**a**) and 20× (**b**) magnification are shown (scale bar: 50 μM); (**c**) total lipid content after Nile Red evaluation in the hepatic tissue; (**d**) number of lipid droplets in the hepatic tissue; (**e**) area of lipid droplets in the hepatic tissue. The results are expressed as mean ± S.E.M.; ** *p* < 0.01, **** *p* < 0.0001; statistical analysis was performed using Student’s *t* test. CTRL = isocaloric control diet; MCD = methionine- and choline-deficient diet; 6w = six weeks.

**Figure 3 ijms-24-09808-f003:**
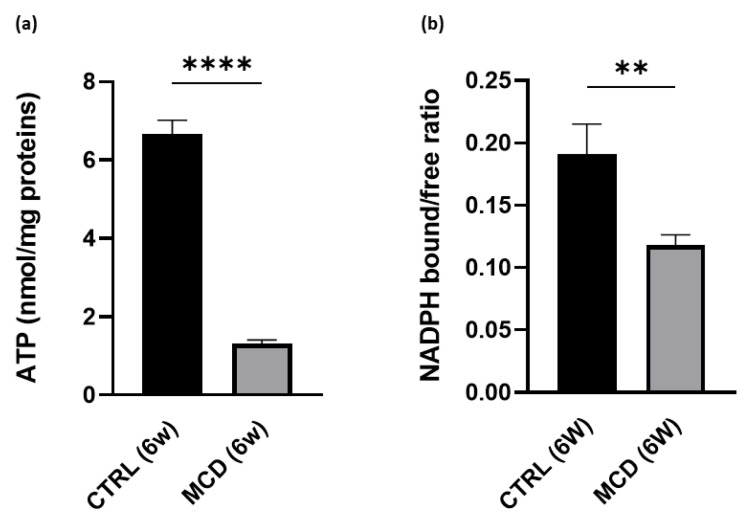
ATP content and NADPH bound/free ratio evaluation in 6-week MCD diet-fed and CTRL rats. The animals were fed with the CTRL (n = 6) and MCD (n = 15) diet for 6 weeks. (**a**) ATP content was evaluated in the hepatic tissue. (**b**) NADPH bound/free ratio was evaluated in the hepatic tissue. The results are expressed as mean ± S.E.M.; ** *p* < 0.01, **** *p* < 0.0001; the statistical analysis was performed using Student’s *t* test. CTRL = isocaloric control diet; MCD = methionine- and choline-deficient diet; 6w = six weeks.

**Figure 4 ijms-24-09808-f004:**
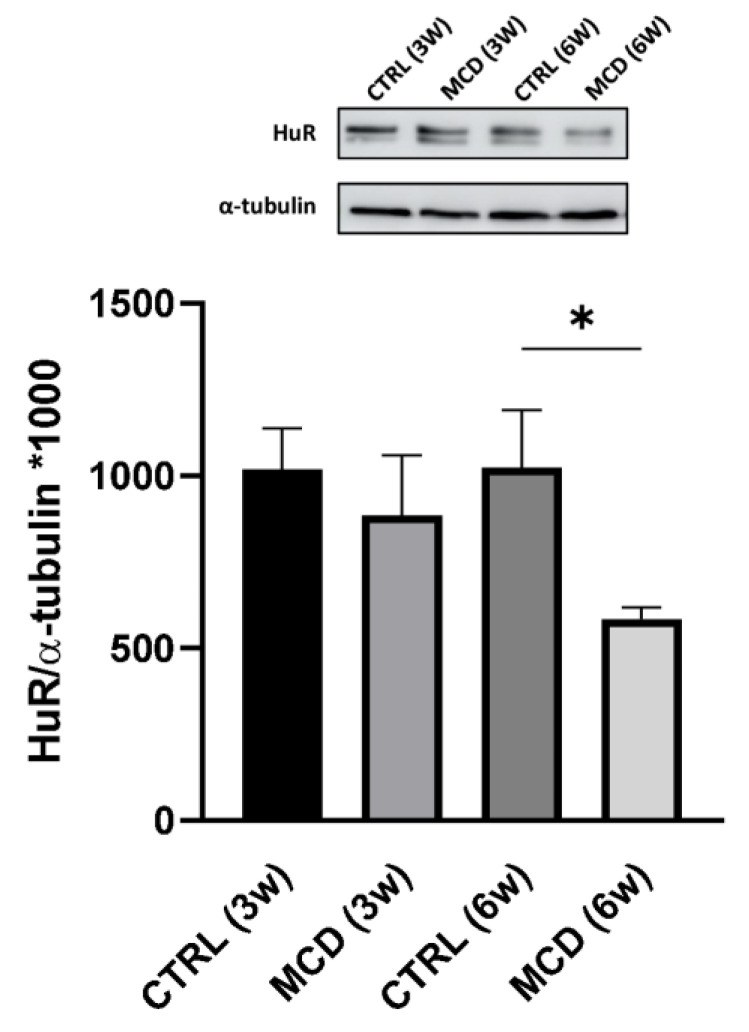
Evaluation of HuR protein expression in 3- and 6-week MCD diet-fed rats and CTRL rats. Upper side: cropped Western blotting images; lower side: histogram of the HuR protein expression in hepatic tissue from rats fed with the CTRL diet for 3 weeks (n = 7) and 6 weeks (n = 6) and the MCD diet for 3 weeks (n = 7) and 6 weeks (n = 15). The results are expressed as mean gray levels ratios × 10^3^ (mean ± S.E.M.) of HuR/α-tubulin immunoreactivities measured by Western blotting; * *p* < 0.05; statistical significance was evaluated using Tukey’s multiple comparison test. CTRL = isocaloric control diet; MCD = methionine- and choline-deficient diet; 3w = three weeks; 6w = six weeks.

**Figure 5 ijms-24-09808-f005:**
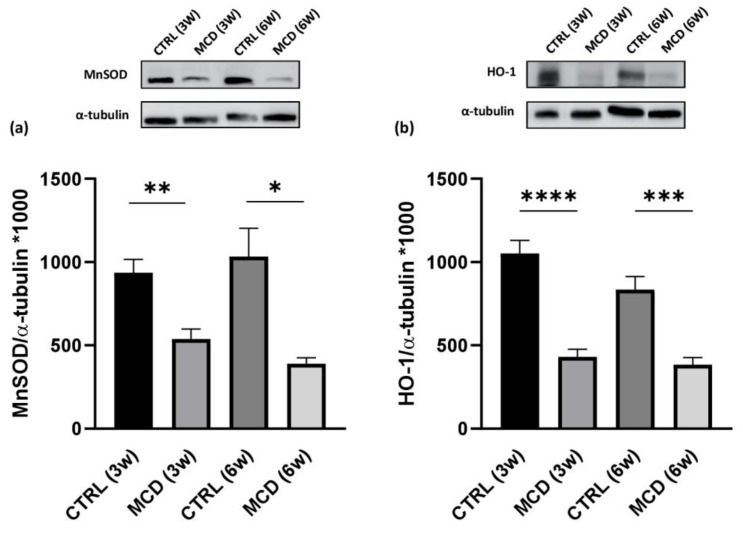
Evaluation of HuR target protein expressions in 3- and 6-week MCD diet-fed rats and CTRL rats. Upper side: cropped Western blotting images; lower side: histograms of MnSOD (**a**) and HO-1 (**b**) protein expression in the hepatic tissue from rats fed with the CTRL diet for 3 weeks (n = 7) and 6 weeks (n = 6) and the MCD diet for 3 weeks (n = 7) and 6 weeks (n = 15). The results are expressed as mean gray levels ratios × 10^3^ (mean ± S.E.M.) of MnSOD/α-tubulin and HO-1/α-tubulin immunoreactivities measured by Western blotting; * *p* < 0.05, ** *p* < 0.01, *** *p* < 0.001, **** *p* < 0.0001; the statistical analysis was performed using the Games–Howell test. CTRL = isocaloric control diet; MCD = methionine- and choline-deficient diet; 3w = three weeks; 6w = six weeks.

**Figure 6 ijms-24-09808-f006:**
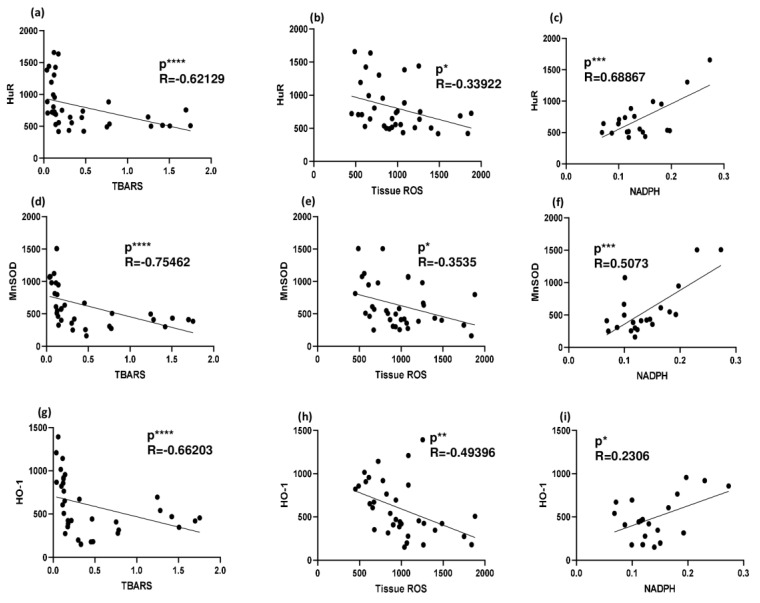
Correlation between HuR (and its targets) and oxidative stress indicators in 3- and 6-week MCD diet and CTRL-fed rats. (**a**) Correlation between HuR and TBARS in rat livers; (**b**) correlation between HuR and ROS production in rat liver; (**c**) correlation between HuR and NADPH in rat liver; (**d**) correlation between MnSOD and TBARS in rat livers; (**e**) correlation between MnSOD and ROS production in rat liver; (**f**) correlation between MnSOD and NADPH in rat liver; (**g**) correlation between HO-1 and TBARS in rat livers; (**h**) correlation between HO-1 and ROS production in rat liver; (**i**) correlation between HO-1 and NADPH in rat liver; * *p* < 0.05, ** *p* < 0.01, *** *p* < 0.001, **** *p* < 0.0001. Pearson’s and Spearman’s tests were used to evaluate statistical significance for parametric and non-parametric data, respectively. CTRL = isocaloric control diet; MCD = methionine- and choline-deficient diet.

**Table 1 ijms-24-09808-t001:** Enzyme release after MCD diet administration.

	3w		6w	
	CTRL	MCD	CTRL	MCD
AST (mU/mL)	84.5 *±* 1.7	109.5 *±* 7.5 *	73.6 *±* 13.7	144 *±* 17.4 *
ALT (mU/mL)	21.5 *±* 2	81.5 *±* 15 *	37.3 *±* 7.6	168.6 *±* 9.8 *

Rats were fed with the CTRL diet for 3 weeks (n = 7) and 6 weeks (n = 6) and with the MCD diet for 3 weeks (n = 7) and 6 weeks (n = 15). The results are expressed as mean ± S.E.M.; * *p* < 0.05. Statistical significance was evaluated with the Tukey test. CTRL = isocaloric control diet; MCD = methionine- and choline-deficient diet; 3w = three weeks; 6w = six weeks.

## Data Availability

The data presented in this study are available on request from the corresponding author.

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
