# Peer review of "MCD Diet Modulates HuR and Oxidative Stress-Related HuR Targets in Rats"

_ijms, 2023, doi:10.3390/ijms24129808_

Round 1

Reviewer 1 Report

The authors checked the expression of HuR, MnSOD, and HO-1 in a rat model of NAFL/NASH induced by methionine-choline deficient (MCD).

The MCD diet caused oxidative stress, mitochondrial dysfunction, and fat accumulation, and also attenuated the expression of HuR.

The paper is well written and the experiments presented seem to be well done.

However, there are many issues that need to be addressed, such as what role HuR really plays as a molecule, for example, overexpression of HuR so that it does not decrease even with MCT diet, or how HuR is different from NASH when it is decreased by molecular methods.

In the Discussion, please describe the limitation of this study and how this study will be specifically developed.

Author Response

Reviewer 1’s comments and suggestions for the authors:

The authors checked the expression of HuR, MnSOD, and HO-1 in a rat model of NAFL/NASH induced by methionine-choline deficient (MCD).

The MCD diet caused oxidative stress, mitochondrial dysfunction, and fat accumulation, and also attenuated the expression of HuR.

The paper is well written and the experiments presented seem to be well done.

However, there are many issues that need to be addressed, such as what role HuR really plays as a molecule, for example, overexpression of HuR so that it does not decrease even with MCT diet, or how HuR is different from NASH when it is decreased by molecular methods.

In the Discussion, please describe the limitation of this study and how this study will be specifically developed.

Authors’ answer:

We thank the reviewer for his positive remarks. We also believs, as the reviewer suggested, that the effects of HuR overexpression or inhibition on NAFLD development should be assessed in our model; we addressed this issue by adding a paragraph in the discussion, describing the limitations of this work and its future developments, as recommended by the reviewer.

Reviewer 2 Report

The manuscript submitted by Ferrigno et al., is an interesting in vivo study investigating the effect of diet on the modulation of enzymes involved in NAFLD. This is an interesting study with potential for clinical implications. 

The reviewer would like to offer below some points for consideration by the authors.

1. Consider describing the rationale for the number of rats used in the study (power calculation etc).

2. The NAFLD is a complex phenomenon whereby several parameters need to be taken into consideration. While the targets identified are important they are clearly not the only ones that influence the outcome as per the disease (manifestation and progression). It is important for such a discussion to be included in the discussion section. 

3. Also, it would be important to expand somewhat including other models in relation to diet and NAFLD in the discussion section. Here is a manuscript that may be helpful in that regard: 

Spooner, H.C.; Derrick, S.A.; Maj, M.; Manjarín, R.; Hernandez, G.V.; Tailor, D.S.; Bastani, P.S.; Fanter, R.K.; Fiorotto, M.L.; Burrin, D.G.; La Frano, M.R.; Sikalidis, A.K.; Blank, J.M. High-Fructose, High-Fat Diet Alters Muscle Composition and Fuel Utilization in a Juvenile Iberian Pig Model of Non-Alcoholic Fatty Liver Disease. Nutrients 2021, 13, 4195. https://doi.org/10.3390/nu13124195.

English language seems OK proofreading is suggested for typos and improvement of flow.

Author Response

Reviewer 2’s comments and suggestions for the authors:

We thank you the reviewer for his positive comments. We answered each remark separately, reviewer’s suggestions are reported below as well as authors’ answers.

  1. Consider describing the rationale for the number of rats used in the study (power calculation etc).

Authors’ answer

We described the power calculation performed when presenting the project to the ethical committee in the Materials and Methods section.

  1. The NAFLD is a complex phenomenon whereby several parameters need to be taken into consideration. While the targets identified are important they are clearly not the only ones that influence the outcome as per the disease (manifestation and progression). It is important for such a discussion to be included in the discussion section.

Authors’ answer

As recommended, we added in the discussion a paragraph about other factors involved in NAFLD development, especially focusing on those that have been demonstrated to be HuR-dependent.

  1. Also, it would be important to expand somewhat including other models in relation to diet and NAFLD in the discussion section. Here is a manuscript that may be helpful in that regard: Spooner, H.C.; Derrick, S.A.; Maj, M.; Manjarín, R.; Hernandez, G.V.; Tailor, D.S.; Bastani, P.S.; Fanter, R.K.; Fiorotto, M.L.; Burrin, D.G.; La Frano, M.R.; Sikalidis, A.K.; Blank, J.M. High-Fructose, High-Fat Diet Alters Muscle Composition and Fuel Utilization in a Juvenile Iberian Pig Model of Non-Alcoholic Fatty Liver Disease. Nutrients 2021, 13, 4195. https://doi.org/10.3390/nu13124195.

Authors’ answer

We thank the reviewer for recommending us this interesting article that had escaped our attention. As suggested, we added to the discussion two other relevant nutritional models: high-fat high fructose diet, as discussed by Spooner et al, 2021, and Cholesterol cholate diet.

  1. English language seems OK proofreading is suggested for typos and improvement of flow.

Authors’ answer

The manuscript was thoroughly revised with the aim of improving the flow and correcting typos.

Round 2

Reviewer 2 Report

The authors have made a reasonable effort in addressing reviewer's comments. Proofreading is suggested.